# FAST CONSTRAINED SAMPLING IN PRE-TRAINED DIFFUSION MODELS

## ABSTRACT

Diffusion models have dominated the field of large, generative image models, with the prime examples of Stable Diffusion and DALL-E 3 being widely adopted. These models have been trained to perform text-conditioned generation on vast numbers of image-caption pairs and as a byproduct, have acquired general knowledge about natural image statistics. However, when confronted with the task of constrained sampling, e.g. generating the right half of an image conditioned on the known left half, applying these models is a delicate and slow process, with previously proposed algorithms relying on expensive iterative operations that are usually orders of magnitude slower than text-based inference. This is counterintuitive, as image-conditioned generation should rely less on the difficult-to-learn semantic knowledge that links captions and imagery, and should instead be achievable by lower-level correlations among image pixels. In practice, inverse models are trained or tuned separately for each inverse problem, e.g. by providing parts of images during training as an additional condition, to allow their application in realistic settings. However, we argue that this is not necessary and propose an algorithm for fast-constrained sampling in large pre-trained diffusion models (Stable Diffusion) that requires no expensive backpropagation operations through the model and produces results comparable even to the state-of-the-art *tuned* models. Our method is based on a novel optimization perspective to sampling under constraints and employs a numerical approximation to the expensive gradients, previously computed using backpropagation, incurring significant speed-ups.

## 1 INTRODUCTION

The recent state-of-the-art in image generation has been dominated by diffusion-based, text-to-image models (Rombach et al., 2022), which excel in translating input text prompts into images. However, by training on millions of text-image pairs, these models have acquired some general knowledge about the natural image space. Thus, besides from text-to-image generation, these models should also be useful in other image-based inference tasks, such as inpainting, super-resolution etc., given that their knowledge about image statistics can be induced.

Such efforts on utilizing text-to-image models on image-based inference tasks have focused on devising ways to adapt pre-trained, text-guided models to the target task. The simplest approach is to fine-tune the text-based model on the new image-based task (Xie et al., 2023; Wang et al., 2024), with the downside of fine-tuning being unnecessarily expensive in smaller-scale scenarios. Alternatively, a different set of algorithms has been developed that utilize only the pre-trained text-to-image diffusion model, and modify the sampling process to infer the missing information (Chung et al., 2023; Rout et al., 2023; Chung et al., 2024). These sampling-based approaches however come with a significant increase in the computations needed.

As a practical example, in Figure 1 we select an image from the web and pose the task of inferring the right half given the pixels in the left half. Based on the Stable Diffusion 1.5 model, we first utilize a fine-tuned variant, trained to be conditioned on images and inpainting masks. Although the inference is fast (4s) and relatively accurate with textures being correctly replicated on the right side and no visible seams, this model required 440k steps of additional inpainting training, on top of the initial SD 1.5 weights.

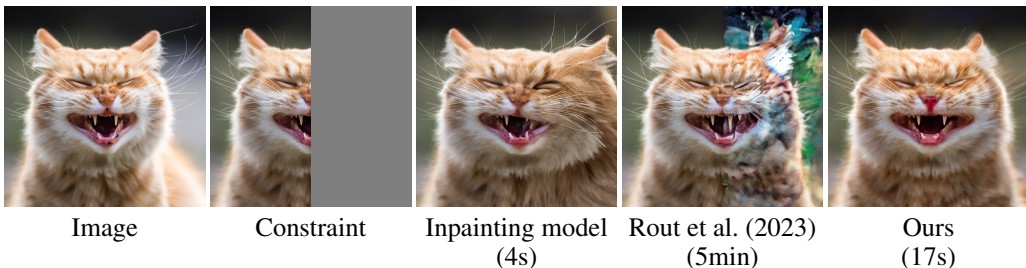

| Image | Constraint | Inpainting model (4s) | Rout et al. (2023) (5min) | Ours (17s) |

Figure 1: Half-image inpainting. Our method allows for fast and accurate inference of the missing part of the image using the original pre-trained Stable Diffusion 1.5 model, which was not tuned for this purpose. In contrast, previous inverse problem solvers are slow and fail to understand the long-range correlations between the image pixels, or rely on models trained/tuned for each given problem. The inpainting model used is the Stable Diffusion 1.5-inpainting fine-tuned model.

Secondly, we demonstrate the inpainting results of a recent sampling-based method, PSLD (Rout et al., 2023). This algorithm requires backpropagating through the denoiser network multiple times during inference, which increases the inference time to 5 minutes. Apart from speed, the method fails to generate a realistic right half, with the generated image achieving an overall correct shape but failing to use the right textures on the fur.

With the shortcomings of existing approaches in mind, we propose a new method for sampling from pre-trained diffusion models under constraints which aims to provide both fast inference speeds and high-quality results. We approach inference under a diffusion prior and a constraint from an optimization perspective and find that (a) there is an alternative gradient update to the diffusion latents during sampling that does not coincide with the gradient updates of previous approaches (b) there is a fast numerical approximation that speeds up this optimization significantly. Our work aims to make inference under constraints practical both by improving the synthesized images and reducing the inference times to a reasonable range.

## 2 BACKGROUND

### 2.1 DENOISING DIFFUSION

Denoising diffusion models were proposed for image generation in Ho et al. (2020), where their ability to generate diverse and high-quality samples was first showcased in the image space. The original formulation views the training and inference process as a hierarchical latent variable model $x_T \to x_{T-1} \to \cdots \to x_1 \to x_0$, where the final latent is distributed normally $x_T \sim N(\mathbf{0}, \mathbf{I})$ and $p(x_0)$ represents the data distribution. After choosing a noise schedule $a_t$ that defines the *forward* transitions $x_t \to x_{t+1}$, usually Gaussian centered at $\sqrt{\frac{a_{t+1}}{a_t}} x_t$ and with variance $(1 - \frac{a_{t+1}}{a_t})$, the model is trained to reverse each individual step in the diffusion process.

Further iterations of denoising diffusion introduced class conditioning (Nichol & Dhariwal, 2021) and classifier-free guidance (Ho & Salimans, 2022), which culminated in the development of Latent Diffusion Models (Rombach et al., 2022). Latent diffusion proposed the large-scale training of text-to-image diffusion models in the latent space of an image autoencoder, $x_0 = \mathcal{D}(\mathcal{E}(x_0))$. We note that although we use latent diffusion models for all our experiments, we do not require additional steps during inference to ensure consistency between the pixel and latent spaces, as is performed in previous works (Rout et al., 2023; Chung et al., 2024).

### 2.2 GRADIENT DESCENT STEPS THROUGH BACKPROPAGATION

A number of recent approaches to solving inverse problems in denoising diffusion models have been investigated in hope that pre-trained large models, which required significant investment, can be used for inference directly, without additional tuning for each different inverse problem (Chung et al., 2023; Rout et al., 2023; Chung et al., 2024). The most typical problem formulation is the denoising generation of the sequence $x_T, x_{T-1}, ..., x_1, x_0$ under the constraint on the final signal in

the form $\boldsymbol{A}\boldsymbol{x}_0 = \boldsymbol{y}$, or using a relaxed version that requires minimization of $||\boldsymbol{A}\boldsymbol{x}_0 - \boldsymbol{y}||_2^2$, possibly as part of the likelihood function $p(\boldsymbol{y}|\boldsymbol{x}_0) = \mathcal{N}(\boldsymbol{y}; \boldsymbol{A}\boldsymbol{x}_0, \sigma^2\boldsymbol{I})$. By Tweedie's formula (Efron, 2011), denoising diffusion models approximating $\nabla_{\boldsymbol{x}_t}\log p_t(\boldsymbol{x}_t)$ can be used to express the expected value of $\boldsymbol{x}_0$, denoted as $\hat{\boldsymbol{x}}_0$, but the addition of the constraint as if an additional observed variable $\boldsymbol{y}$ was generated requires the addition of the term $\nabla_{\boldsymbol{x}_t}\log p(\boldsymbol{y}|\boldsymbol{x}_0)$.

At any rate, the regular denoising diffusion steps are altered so that at each $t$, the generated latent $\boldsymbol{x}_t$ is moved in the direction reducing the cost

$$C(\boldsymbol{x}_t) = (\boldsymbol{A}\hat{\boldsymbol{x}}_0(\boldsymbol{x}_t) - \boldsymbol{y})^T(\boldsymbol{A}\hat{\boldsymbol{x}}_0(\boldsymbol{x}_t) - \boldsymbol{y}). \tag{1}$$

For example, in the case of inpainting applications, matrix $\boldsymbol{A}$ extracts a subsection of the pixels in image $\boldsymbol{x}_0$ to be compared with a given target $\boldsymbol{y}$. The estimated expected value of $\boldsymbol{x}_0$ at the end of the chain is provided by the diffusion model as a nonlinear function $\hat{\boldsymbol{x}}_0(\boldsymbol{x}_t)$. Typically, these moves are gradient descent moves, i.e. moves of $\boldsymbol{x}_t$ in the direction

$$\boldsymbol{h} = -\nabla_{\boldsymbol{x}_t}C(\boldsymbol{x}_t) = -\boldsymbol{J}^T\boldsymbol{A}^T(\boldsymbol{A}\hat{\boldsymbol{x}}_0 - \boldsymbol{y}) = -\boldsymbol{J}^T\boldsymbol{e}, \quad \boldsymbol{J} = \nabla_{\boldsymbol{x}_t}\hat{\boldsymbol{x}}_0(\boldsymbol{x}_t), \ \boldsymbol{e} = \boldsymbol{A}^T(\boldsymbol{A}\hat{\boldsymbol{x}}_0 - \boldsymbol{y}) \tag{2}$$

Computation of the Jacobian $\boldsymbol{J}$ would be expensive both in memory and computation, so the gradient is computed using backpropagation through $C(\boldsymbol{x}_t)$, and we show its mathematical form here for comparison with a different update we use in this paper. In Chung et al. (2023) and Chung et al. (2024) for example, (several) gradient steps of this form are applied to $\boldsymbol{x}_t$ at each step $t$ of generation before moving on to the next stage. As optimization of $\boldsymbol{x}_t$ might reduce the total noise in the image below what the denoising at $t-1$ was trained for, the gradient steps moving $\boldsymbol{x}_t$ towards optimizing $C(\boldsymbol{x}_t)$ are combined with adding additional noise, which could also be seen as a form of stochastic averaging.

## 3 OUR APPROACH: NEWTON STEPS BASED ON THE INVERSE FUNCTION

Suppose that at the current point in denoising, $(\boldsymbol{x}_t, \hat{\boldsymbol{x}}_0)$, the function $\hat{\boldsymbol{x}}_0(\boldsymbol{x}_t)$ is locally invertible, i.e. there is a unique, although unknown, inverse function $\boldsymbol{x}_t(\hat{\boldsymbol{x}}_0 + \boldsymbol{g})$, in the neighborhood of $\hat{\boldsymbol{x}}_0$ (for small $||\boldsymbol{g}||$), in which case $\nabla_{\hat{\boldsymbol{x}}_0}\boldsymbol{x}_t = (\nabla_{\boldsymbol{x}_t}\boldsymbol{x}_0)^{-1} = \boldsymbol{J}^{-1}$. This assumption may seem obviously wrong considering that denoising diffusion models are trained assuming that $\boldsymbol{x}_t$ is distributed as a Gaussian variable centered at $\boldsymbol{x}_0$, i.e. for a given $\boldsymbol{x}_0$ from the data distribution, many different $\boldsymbol{x}_t$ samples are feasible. However, $\hat{\boldsymbol{x}}_0$ is a deterministic function represented by a neural network, through which backpropagation can compute the update in (2). I.e., at least for the directions we are interested in, unless the cost is already optimized, the perturbations in $\hat{\boldsymbol{x}}_0$ will yield nonzero perturbations in $\boldsymbol{x}_t$.

Using the first-order Taylor series we can approximate

$$\boldsymbol{x}_t(\hat{\boldsymbol{x}}_0 + \boldsymbol{g}) \approx \boldsymbol{x}_t(\hat{\boldsymbol{x}}_0) + (\nabla_{\hat{\boldsymbol{x}}_0}\boldsymbol{x}_t)\boldsymbol{g} = \boldsymbol{x}_t + \boldsymbol{J}^{-1}\boldsymbol{g} \tag{3}$$

for some small perturbation vector $\boldsymbol{g}$.

Now suppose that, as in the Gauss-Newton method, $\boldsymbol{x}_t$ should move as close as possible to some target $\boldsymbol{x}_t'$, i.e. we need to minimize

$$C(\hat{\boldsymbol{x}}_0 + \boldsymbol{g}) = (\boldsymbol{x}_t(\hat{\boldsymbol{x}}_0 + \boldsymbol{g}) - \boldsymbol{x}_t')^T(\boldsymbol{x}_t(\hat{\boldsymbol{x}}_0 + \boldsymbol{g}) - \boldsymbol{x}_t') \approx (\boldsymbol{x}_t + \boldsymbol{J}^{-1}\boldsymbol{g} - \boldsymbol{x}_t')^T(\boldsymbol{x}_t + \boldsymbol{J}^{-1}\boldsymbol{g} - \boldsymbol{x}_t') \tag{4}$$

w.r.t. to the movement $\boldsymbol{g}$ of $\hat{\boldsymbol{x}}_0$. Setting $\nabla_{\boldsymbol{g}}C(\hat{\boldsymbol{x}}_0 + \boldsymbol{g}) = \boldsymbol{0}$ yields the system

$$(\boldsymbol{J}^{-1})^T(\boldsymbol{x}_t' - \boldsymbol{x}_t) = (\boldsymbol{J}^{-1})^T(\boldsymbol{J}^{-1})\boldsymbol{g} \tag{5}$$

$$\boldsymbol{x}_t' - \boldsymbol{x}_t = \boldsymbol{h} = \boldsymbol{J}\boldsymbol{g} \tag{6}$$

Thus, locally, within the validity of the first order approximation, $\boldsymbol{g}$ and $\boldsymbol{h}$ represent a pair of corresponding movements in $\hat{\boldsymbol{x}}_0$ and $\boldsymbol{x}_t$ respectively, but here they are linked differently than in (2). The semantics of the relationship here is that $\boldsymbol{g}$ is the best direction in $\hat{\boldsymbol{x}}_0$ to approximate the target move $\boldsymbol{h}$ in $\boldsymbol{x}_t$ in the least squares sense. If we assume that $\boldsymbol{g} = -\epsilon\boldsymbol{e}$, where $\boldsymbol{e}$ is the error vector on $\hat{\boldsymbol{x}}_0$ as before in (2) and $\epsilon$ is a small constant used to make the first order approximation valid, we get

$$\boldsymbol{h} = -\epsilon\boldsymbol{J}\boldsymbol{e}, \tag{7}$$

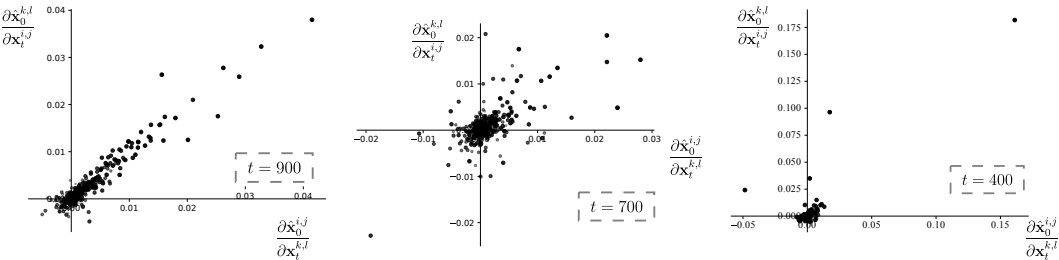

Figure 2: Sample pairs of pixels $(i, j), (k, l)$ of the denoiser Jacobian $\nabla_{\boldsymbol{x}_t} \hat{\boldsymbol{x}}_0 = \nabla_{\boldsymbol{x}_t} E[\boldsymbol{x}_0 | \boldsymbol{x}_t]$ for different timesteps $t$. We use Stable Diffusion 1.5.

and so moving $\boldsymbol{x}_t$ in this direction would mean moving it to the target $\boldsymbol{x}_t'$ that would *require* movement of $\hat{\boldsymbol{x}}_0$ in the direction opposite of the error vector $\boldsymbol{e}$. For example, in the case of image inpainting, the error vector (per pixel) $\boldsymbol{e} = \boldsymbol{A}^T(\boldsymbol{A}\hat{\boldsymbol{x}}_0 - \boldsymbol{y})$ has zeros for pixels outside of the given (constraint) area. So the move $\boldsymbol{h}$ in $\boldsymbol{x}_t$ has to be such so that by searching for a compensatory move in $\boldsymbol{g}$ in $\hat{\boldsymbol{x}}_0$ the resulting move would not alter $\hat{\boldsymbol{x}}_0$ in the areas outside of the constraint.

The direction of optimization we propose in (5) has at least two advantages over the usual gradient descent update. First, the direction $\boldsymbol{h}$ can be computed numerically to save both on computation (2-fold) and memory (2.5-fold) compared to using backpropagation on cost (1). To derive that update, consider the function $f(s) = \hat{\boldsymbol{x}}_0(\boldsymbol{x}_t - s\boldsymbol{e})$ where the variable $s$ is scalar. Its derivative at $s = 0$ is

$$\frac{df}{ds} = -\boldsymbol{J}\boldsymbol{e}, \tag{8}$$

and so the direction $\boldsymbol{h}$ can be approximated numerically using the approximate derivative of $f(s)$ at $s = 0$

$$f' = \frac{1}{\delta}[\hat{\boldsymbol{x}}_0(\boldsymbol{x}_t + \delta\boldsymbol{e}) - \hat{\boldsymbol{x}}_0(\boldsymbol{x}_t)], \tag{9}$$

$$\boldsymbol{h} = -\epsilon\boldsymbol{J}\boldsymbol{e} \approx \frac{\epsilon}{\delta}[\hat{\boldsymbol{x}}_0(\boldsymbol{x}_t + \delta\boldsymbol{e}) - \hat{\boldsymbol{x}}_0(\boldsymbol{x}_t)], \tag{10}$$

requiring no backpropagation but instead, two forward passes through the network: one to compute $\hat{\boldsymbol{x}}_0(\boldsymbol{x}_t)$ and another to compute $\hat{\boldsymbol{x}}_0$ for the perturbed $\boldsymbol{x}_t$ (in the negative direction of the error vector).

The second advantage comes from the asymmetry of the Jacobian $\boldsymbol{J}$, as alluded to above. The updates (2) and (5) optimize locally for different things unless $\boldsymbol{J} = \boldsymbol{J}^T$. As described in 3.2 on inpainting, (5) is in the direction that would represent the textures in the constrained region everywhere in the rest of the image as much as possible, whereas (5) only leads to copying some of the texture into some parts of the rest of the image.

There is no theoretical reason for the Jacobian $\boldsymbol{J}$ to be asymmetric in pre-trained denoising diffusion models in general. In fact, it very well could be symmetric as these models can be seen in terms of score matching so that $\hat{\boldsymbol{x}}_0$ approximates the true expectation $E[\boldsymbol{x}_0 | \boldsymbol{x}_t] = \frac{1}{\sqrt{\alpha_t}}[\boldsymbol{x}_t + \boldsymbol{v}_t \nabla_{\boldsymbol{x}_t} \log p_t(\boldsymbol{x}_t)]$, and the gradient of this is indeed symmetric:

$$\nabla_{\boldsymbol{x}_t} E[\boldsymbol{x}_0 | \boldsymbol{x}_t] = \frac{1}{\sqrt{\alpha_t}}[\boldsymbol{I} + \boldsymbol{v}_t \nabla^2 \log p_t(\boldsymbol{x}_t)]. \tag{11}$$

One could argue that a well-trained model $\hat{\boldsymbol{x}}_0(\boldsymbol{x}_t)$ should have its Jacobian equal to this. But whether the denoising diffusion models are trained with score matching in mind or using the variational method of Ho et al. (2020), they do not directly optimize to match the real score $\nabla_{\boldsymbol{x}_t} \log p_t(\boldsymbol{x}_t)$ everywhere nor are they constrained to produce symmetric Jacobians, and as shown in 3.1 the Jacobian for Stable Diffusion 1.5 is not symmetric, which we assume is the reasons why our optimization of $\boldsymbol{x}_t$ may be more suitable for some applications, as demonstrated in Section 4.

## 3.1 SYMMETRY OF THE JACOBIAN

In Figure 2 we verify the claim that the trained diffusion model's Jacobian is not symmetric. We again employ SD 1.5 and given a random input image from ImageNet, we scale it and add noise

| Sample | Condition | $\mathbf{J}^{\mathrm{T}}\mathbf{e}$ | $\mathbf{J}\mathbf{e}$ |

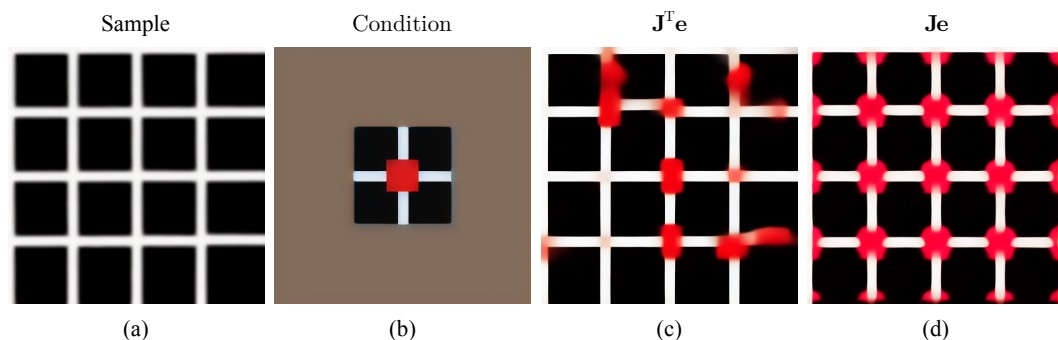

| (a) | (b) | (c) | (d) |

Figure 3: Comparison between 5 gradient updates at $t = 800$ using a learning rate $\lambda = 1$.

---

**Algorithm 1** The proposed algorithm for linear inverse problem solving.

**Input:** Pre-trained diffusion model $\hat{\boldsymbol{x}}_0(\boldsymbol{x}_t)$, diffusion schedule parameters $\zeta_t, \kappa_t, \beta_t$, operator $\boldsymbol{A}$, measurement $\boldsymbol{y}$, step size $\delta$, optimization iterations $K$, learning rate $\lambda$
$\boldsymbol{x}_T \sim N(\boldsymbol{0}, \boldsymbol{I})$
**for** $t \in \{T, T - s, T - 2s, \ldots, s\}$ **do**
    **while** $i < K$ **do**
        $\boldsymbol{e} = \boldsymbol{A}^T(\boldsymbol{A}\hat{\boldsymbol{x}}_0(\boldsymbol{x}_t) - \boldsymbol{y})$                                        ▷ Depends on the task
        $\boldsymbol{h} = [\hat{\boldsymbol{x}}_0(\boldsymbol{x}_t + \delta\boldsymbol{e}) - \hat{\boldsymbol{x}}_0(\boldsymbol{x}_t)]/\delta$
        $\boldsymbol{x}_t = \boldsymbol{x}_t + \lambda\boldsymbol{h}$
        $i = i + 1$
    **end while**
    $\boldsymbol{z}_t \sim N(\boldsymbol{0}, \boldsymbol{I})$
    $\boldsymbol{x}_{t-s} = \zeta_t\boldsymbol{x}_t + \kappa_t\hat{\boldsymbol{x}}_0(\boldsymbol{x}_t) + \beta_t\boldsymbol{z}_t$            ▷ Step using DDIM (Song et al., 2020) [1]
**end for**
**Return:** $\boldsymbol{x}_0$

---

to get an intermediate latent of the diffusion process at different timesteps. We then pass the image through the denoiser and compute the gradients $\partial\hat{\boldsymbol{x}}_0^{k,l}/\partial\boldsymbol{x}_t^{i,j}$ and $\partial\hat{\boldsymbol{x}}_0^{i,j}/\partial\boldsymbol{x}_t^{k,l}$ for randomly chosen pixels $(i, j), (k, l)$ using backpropagation. When plotting the gradients we see that the values deviate from $y = x$, which would indicate a symmetric Jacobian.

### 3.2 DIFFERENCE BETWEEN GRADIENT DIRECTIONS

To highlight the difference between the two gradient updates of (2) and (5) we perform the experiment showcased in Figure 3. We create a synthetic black-and-white grid image (Figure 3 (a)), which we will pass to the diffusion model. We compute the diffusion latent at $t = 800$ by first blurring and scaling the original image and then adding the appropriate magnitude noise. Then we aim to find how the model intends to change the entire image when asked to add a red square in the middle. We pose this as an inpainting task and compute the two different gradients when given a measurement corresponding to the center patch of the image with an added red square (Figure 3 (b)). Starting from the initial diffusion latent, we perform 5 gradient updates using the two different gradients and visualize the final $\hat{\boldsymbol{x}}_0$ produced by the model.

The resulting images show that the proposed direction (Figure 3 (d)) produces a more coherent image that copies the newly introduced texture to the correct locations (intersections). The direction given by backpropagation (Figure 3 (c))

---

[1]The diffusion schedule parameters $\zeta_t, \kappa_t$ that we use in our notation can be trivially computed from the schedule variances $a_t$ as described in DDIM.

| Method | Inpaint (Freeform) | | | SR ($\times 8$) | | | Time |
|---|---|---|---|---|---|---|---|
| | PSNR↑ | LPIPS↓ | FID↓ | PSNR↑ | LPIPS↓ | FID↓ | (approx.) |
| Ours | 22.20 | 0.275 | 30.45 | 22.29 | 0.428 | 73.05 | 2min |
| P2L (Chung et al., 2024) | 21.99 | 0.229 | 32.82 | 23.38 | 0.386 | 51.81 | 30min |
| LDPS | 21.54 | 0.332 | 46.72 | 23.21 | 0.475 | 61.09 | 8min |
| PSLD (Rout et al., 2023) | 20.92 | 0.251 | 40.57 | 23.17 | 0.471 | 60.81 | 12min |

Table 1: Quantitative evaluation (PSNR, LPIPS, FID) of free-form inpainting and superresolution ImageNet.

## 4 EXPERIMENTS

### 4.1 INVERSE PROBLEMS

We validate our approach on ImageNet (Deng et al., 2009) by performing inpainting and super-resolution. Following Chung et al. (2024), we randomly choose 1000 images from the 10k test images of the `ctest10k` split. For the diffusion model, we use SD v1.5, which is pre-trained on the LAION (Schuhmann et al., 2022) text-image dataset. All experiments were done on a single NVIDIA RTX A5000 24GB GPU.

We use Stable Diffusion 1.5, a latent diffusion model, which spatially compresses the images by a factor of 8. To demonstrate the versatility of our approach, we approach linear inverse problems from two different angles. For inpainting, instead of decoding the latent into an image and comparing it to the measurement in pixel space, we opt to dilate the pixel-level masks and directly perform inpainting in the latent space. That means we 'discard' some of the information in the given image by only keeping $8 \times 8$ patches that do not overlap with the masked pixels. Consequently, we apply the masking and un-masking operators, $\boldsymbol{A}$ and $\boldsymbol{A}^T$ respectively, in the latent space and inpaint latent values instead of pixels. Even with fewer pixels, our method performs better than existing methods in inferring the missing information. We utilize the $10-20\%$ free-form masking from Saharia et al. (2022) as the method of masking pixels.

For superresolution, we cannot work in the latent space as the operation of downsampling pixels does not correspond to downsampling latents. Therefore, in order to compute the error direction $\boldsymbol{e}$ we backpropagate the pixel-level constraint cost $(\boldsymbol{A}\mathcal{D}(\boldsymbol{x}_t) - \boldsymbol{y})^T(\boldsymbol{A}D(\boldsymbol{x}_t) - \boldsymbol{y})$ through the decoder network and utilize the resulting gradient w.r.t. the latent as the error direction. We note that by setting $\boldsymbol{e} = -\partial C(\mathcal{D}(\boldsymbol{x}_t)/\partial \boldsymbol{x}_t$ we only require backpropagation through the decoder model, which is significantly less expensive than backpropagating through the denoiser network, as is done in previous works. For both degradations, we also include additive white Gaussian noise with $\sigma_{\boldsymbol{y}} = 0.05$.

In both experiments, we also find it useful to perform warm restarts of our algorithm. After running Algorithm 1 from $t = 1000$ to $t = 0$, we reset the inferred $x_0$ by adding the appropriate noise to When superresolving, we add an additional perturbation to the gradient to avoid local minima. We find that our approach quickly converged to blurry images, which satisfy completely the superresolution constraint, but do not contain the desired details and texture. A simple perturbation of the gradient with random noise around the current $\hat{\boldsymbol{x}}_0$ seems to be adequate in solving this issue. For inpainting, this is not a problem, as the textures and overall fidelity of the inferred image are dictated by the given regions.

In Figure 4 we showcase the qualitative performance of our algorithm compared to similar recent works. In the case of inpainting our results show greater coherence with the rest of the image. For superresolution, our method seems to introduce more new textures in the generated image, which may not always align perfectly with the shown content (e.g. artifacts). In Table 1 we present quantitative results by measuring PSNR, LPIPS and FID scores. Inpainting shows a clear advantage for our approach, whereas superresolution struggles to improve significantly. It is important to consider that for our results, the inference time is around 2 minutes, including the warm restarts, whereas competing methods always require more than 4 times longer for a single image.

| Measurement | GT | DPS | LDPS | PSLD | P2L | Ours |
|---|---|---|---|---|---|---|

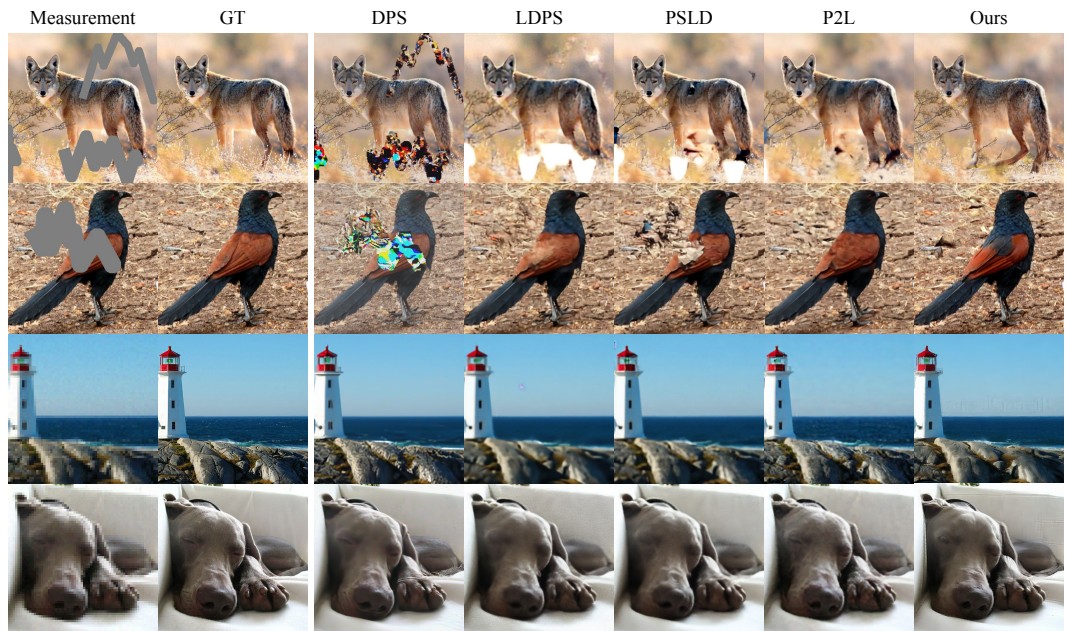

Figure 4: Comparison between our method and other works. We directly use the images and results from (Chung et al., 2024) since there is no code available to replicate their method. We run our algorithm on the same ground truth images for inpainting the given 'gray region' masks and ×8 super-resolution.

## 4.2 IMAGE LAYER INFERENCE

Our approach allows for fast inference of missing regions in a given image. With that in mind, we go beyond inpainting and propose a new inference problem for pre-trained diffusion models, *layer inference*. Given an input image $x_0$, we aim to generate two new images $x_0^1$, $x_0^2$ and a pixel-wise mask $m$ that will satisfy $x_0 = mx_0^1 + (1 - m)x_0^2$. We want the model to generate a possible 'decomposition' of the input image into two layers, e.g. foreground and background along with the blending mask.

We propose a simple algorithm to perform this task. The assumption is that pixels originating from the same layer should have stronger correlations than pixels from different layers. To probe this correlation given a soft mask $m$, we generate multiple possible images by sampling a binary mask from the (0,1) values of $m$ and performing inpainting. After running multiple iterations of inpainting, we can then compute the likelihood of each pixel in the original image to belong in one of the layers. We choose to model a layer image as Gaussian with per-pixel means and variances dictated by the generated samples in that layer.

Instead of performing the full sampling process, we find that we can substitute each layer image with the predicted $\hat{x}_0(x_t)$. To induce more variety to the $x_0$ estimates, we perturb $x_t$ and help generate multiple inpainting variations without running the full inference process. The mask is randomly initialized with values in (0,1) and we choose to sample $K = 5$ images per layer.

In Figure 5 we provide qualitative results by running the layer inference algorithm on images from the web. To guide each layer we also condition the denoiser $\hat{x}_0(x_t)$ on layer-specific text prompts that describe the image content we want to select. The results show that we can control the separation between the layers and get two images that describe different characteristics of the source image. The key is the speed of our method in inferring the missing pixels; this inference would have been computationally infeasible to perform with previous sampling-based approaches.

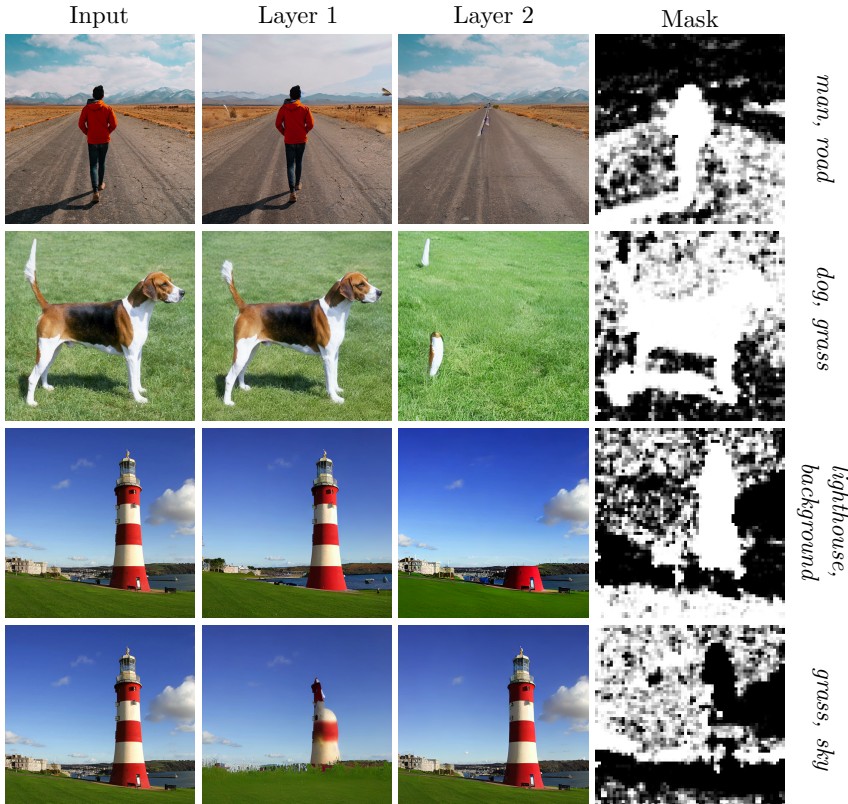

Figure 5: Given an input image we sample two new images which when blended with an inferred mask, reconstruct the original input. Our approach allows us to perform decomposition of random images from the web, guided by simple text prompts.

## 5 CONCLUSION

In this work, we presented a new algorithm for fast inference in pre-trained diffusion models under constraints. Our novel approach exploits a different gradient update that has different qualitative results and requires no expensive backpropagation operations through the model. Our method produces results comparable to the state-of-the-art with significantly less inference time. We also introduce a new layer inference task, which is enabled by our fast and high-quality constrained sampling algorithm. We believe that a fast and accurate method to sample from large pre-trained generative image models, under any condition, can have the potential to enable countless downstream applications that rely on utilizing a strong image prior.

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
