# OpenReview forum: "Fast constrained sampling in pre-trained diffusion models"
_ICLR.cc/2025/Conference — ICLR 2025 Conference Withdrawn Submission_

### Official Review · Reviewer_i3q9 · 2024-10-19

**Soundness:** 2
**Presentation:** 2
**Contribution:** 2
**Rating:** 3
**Confidence:** 4

**Summary:**

The paper examines generative modeling for in-painting and introduces a sampling algorithm for rapid in-painting using pre-trained Stable Diffusion models. Their approach utilizes numerical gradient approximations to accelerate sampling, leading to faster in-painting results in empirical evaluations.

**Strengths:**

This method is ambitious as it aims to replace computationally expensive gradient operations for acceleration.

**Weaknesses:**

* The writing should be more polished and structured for better coherence.

* In experiments, consider conducting additional experiments (Table 1) and ensuring that baseline methods are comprehensive and representative. Since pre-trained Stable Diffusion models were mentioned, it would be beneficial to include evaluations for text-to-image tasks.

**Questions:**

* Figure 1 indicates that the Inpainting model is faster (4s) compared to your method (17s). Does this suggest that the default Inpainting method remains more efficient?

* Regarding soundness, is Formula 1's assumption of linear multiplication form realistic and expressive enough?

* In applying Newton method, does it require second order gradient?

* In Algorithm 1, concerning the calculation of e, what does 'depends on the tasks' imply? Can you provide further elaboration on this?

---

### Official Review · Reviewer_WifF · 2024-11-03

**Soundness:** 2
**Presentation:** 1
**Contribution:** 2
**Rating:** 3
**Confidence:** 3

**Summary:**

This paper proposes a method that mitigates the "heavy" backpropagation of constrained sampling (inverse problem) with diffusion models by first-order approximation to the gradients to use Gauss-Newton methods. By doing this, the paper found that using Jacobian (rather than the Jacobian transpose) works for inverse problem, while these two are different empirically in general inverse problem and constrained sampling settings. The algorithm consists of three steps:
1. Find an error vector $e=A^\top (A \top{x}_0 (x_t)  - y)$, which is the "direction" of error of $x_0$.
2. Based on $e$, find the error vector for noisy $x_t$, $h$.
3. Propagate $x_t$ to the direction of $h$.

**Strengths:**

* By introducing the non-symmetry of Jacobian in the inverse problem setting, solving $h=Jg$ instead of $h=J^\top g$, after proposing some motivating examples on comparing the symmetric Jacobians.
* By excluding the gradient descent in the algorithm, this method achieved faster and memory-efficient solving of diffusion inverse problem, with better performance for inpainting and super-resolution tasks, and enabled the image layer inference.

**Weaknesses:**

* The paper is hard to read, by having lots of typos, grammar errors and unreleased arguments.
* More derivation on the Jacobian computation is required: too much part is omitted.

**Questions:**

* Does Equation (5) and (6) correspond? (5) contains $J^{-1}$ while (6) contains $J$.
* (Line 182) It will be better if additional arguments on the computational costs. (For instance, big-O notation, wall-clock time, FLOPs or memory constraints.)

(Minor typos and grammar check)
* (LaTeX) $\hat{x_0}$ and $\hat{x}_0$ are mixed up in the manuscript. I recommend unifying to $\hat{x}_0$.
* The paragraph in Line 134 is confusing: too many parentheses and division of phrases. For example, \
\
Suppose that at the current point in denoising, $(x_t, \hat{x}_0)$, the function $\hat{x}_0 (x_t)$ is locally invertible, i.e, there is a unique, although unknown, inverse function $x_t(\hat{x}_0 + g)$, in the neighborhood of $\hat{x}_0$ ...
\
can be refined to
\
Suppose that the denoiser function $\hat{x}_0$ is locally invertible over an $\varepsilon$-ball centered at $x_t$ ... \
\
Moreover, this denoising function should have time $t$ as its argument, of sub (or super)-script of the function, for a finer understanding.
* (Line 198) whereas (5) $\to$ whereas (2)

---

In conclusion, this paper is not ready for publication in the current stage of manuscript, mainly because of the difficulty of understanding the writing. Even though the paper seems to contain enough novelty, the manuscript seems to be written in a hurry, yielding a draft rather than the full paper.

---

### Official Review · Reviewer_61L5 · 2024-11-04

**Soundness:** 3
**Presentation:** 3
**Contribution:** 4
**Rating:** 3
**Confidence:** 4

**Summary:**

The paper describes a method for fast constrained sampling in pre-trained diffusion models, such as Stable Diffusion, without the need for expensive backpropagation operations. The key idea is to reformulate the inference process using an optimization perspective that speeds up constrained sampling. This approach focuses on efficiently estimating gradient updates through numerical approximations rather than full backpropagation, significantly reducing the amount of computation.

**Strengths:**

* The paper fairly clearly explains the method. The approach also implements multiple baselines on the problems for infilling. The paper's methodology is rather original and I do not recall seeing it in the literature. The method could help to use diffusion models to solve inverse methods without retraining on task specific tasks or do expensive backprop.

**Weaknesses:**

* Figure 2 is a bit hard to read
* Tasks are somewhat simplistic. Showing the results on more challenging inverse problems aside from infilling would be make the paper stronger
* It would be helpful to using a Deep Image Prior for these or classical infilling methods that do not require task specific training. The convolutional prior of the Stable Diffusion UNet could be doing a lot of heavy lifting. Although this method approach is orthogonal, it could serve as helpful baseline.
* Why isn't GPU memory analyzed here? if no backpropogation is used, I would expect significant memory savings that would allow a much larger batch size and therefore make the speed up vs the baselines even more dramatic for a fixed memory budget in table 1.
* Figure 4.2 is lacking any baselines.

**Questions:**

* The examples show in for the teasers lack creativity and do not demonstrate robustness of the approach. If you really want to show the verstality of the method it would be really enlightening to try to frankenstein two different images together and see if the method is able to create an image on the natural image prior. That is to say if I take an image an outline the left third with a dog head, and the right third with say a cat head, can it connect the images in a convincing way?

* How many objects can the layer inference handle? It would be useful if you could use it to segment multiple objects or part of objects from an image at once.

* FID is evaluated, but it does not clarify how many images it is evaluated on. Note: FID is a biased metric, so it's value will vary depending on how many images you run it on (ie running it on more images will lower the value). If I am reading it correctly, This is only calculated on for $FID_1000$ and $FID_10000$ is usually the smallest value reported in the literature. Otherwise, one should use Kernal Inception Distance (Demystifying MMD GANs) as it's far more reliable on such a small number of samples. I am concerned too because such a small sampling of images from ImageNet will not even cover all the ImageNet classes.

All in all, this paper shows a lot of promise, but I worry the empirical results are entirely qualitative without a sufficient number of examples. The quantitative numbers in Table 1 provide a very mixed bag of results as well.

---

### Official Review · Reviewer_rmiY · 2024-11-04

**Soundness:** 3
**Presentation:** 3
**Contribution:** 2
**Rating:** 6
**Confidence:** 3

**Summary:**

The paper introduces a faster approach for constrained sampling in pre-trained diffusion models. The key innovation lies in the reformulation of the gradient update step: instead of using expensive backpropagation through the model to optimize constraints, the authors derive an alternative update direction by treating the denoiser's output as locally invertible. This enables approximation using just two forward passes, making it faster than traditional methods. The algorithm iteratively updates latent variables at each diffusion timestep using this efficient numerical approximation, achieving comparable or better results than existing methods while being faster. Using Stable Diffusion 1.5, they demonstrate the method's effectiveness on tasks like inpainting and super-resolution.

**Strengths:**

The paper's primary strength lies in its technical contribution that offers a practical solution to a significant problem in diffusion models. By developing a mathematically rigorous alternative to traditional backpropagation-based methods, the authors achieve impressive speed improvements, while maintaining or improving quality. This makes constrained sampling more practical for applications.

Another strength is the method's ability to work directly with pre-trained models without requiring task-specific fine-tuning. The authors validate their approach thoroughly, providing both quantitative metrics and qualitative results across multiple tasks, particularly showing strong performance in inpainting applications. They also ground their practical results in solid theoretical analysis, examining the Jacobian's asymmetry in diffusion models.

The introduction of the "layer inference" task further demonstrates the method's practical value, showing how the faster sampling enables new applications that weren't previously feasible due to computational constraints.

**Weaknesses:**

The paper's main weakness lies in its limited scope of validation. While the method shows strong performance on inpainting tasks, its effectiveness on super-resolution is less impressive. This suggests potential limitations in the method's general applicability that are not fully explored or explained.

From a theoretical perspective, while the authors provide interesting analysis of the Jacobian's asymmetry, they do not fully explain why this leads to better results in practice, leaving some theoretical questions unanswered. Additionally, the lack of direct comparisons for their novel layer inference task makes it difficult to assess the relative improvement over potential baseline approaches.

**Questions:**

While the method shows promising results on inpainting and super-resolution tasks, how well would it perform for broader inverse problems?

What are the failure cases of this approach, and how sensitive is it to hyperparameter choices?

**Details Of Ethics Concerns:**

No concern.

---

### Note · Authors · 2024-11-26

I have read and agree with the venue's withdrawal policy on behalf of myself and my co-authors.